# Insights into the Flavor Differentiation between Two Wild Edible *Boletus* Species through Metabolomic and Transcriptomic Analyses

**DOI:** 10.3390/foods12142728

**Published:** 2023-07-18

**Authors:** Kaixiang Chao, Tuo Qi, Qionglian Wan, Tao Li

**Affiliations:** 1School of Chemistry Biology and Environment, Yuxi Normal University, Yuxi 653100, China; ckx@yxnu.edu.cn (K.C.); wanqionglian@yxnu.edu.cn (Q.W.); 2Ecological Security and Protection Key Laboratory of Sichuan Province, Mianyang Normal University, Mianyang 621000, China; qituo777@163.com

**Keywords:** mushroom flavor, wild edible *Boletus*, UHPLC-MS, metabolome, transcriptome

## Abstract

Despite the popularity of wild edible mushrooms due to their delectable flavor and nutritional value, the mechanisms involved in regulating and altering their taste remain underexplored. In this study, we analyzed the metabolome and transcriptome of *Boletus brunneissimus (B. brunneissimus) and Leccinum extremiorientale (L. extremiorientale)*, two *Boletus* species collected from different environments. Using UHPLC-MS, we annotated 644 peaks and identified 47 differential metabolites via OPLS-DA analysis. Eight of these were related to flavor, including L-Aspartic acid, Glycine, D-Serine, L-Serine, L-Histidine, Tryptophan, L-Isoleucine, Isoleucine, and alpha-D-Glucose. These differential metabolites were mainly concentrated in amino acid metabolism pathways. Transcriptome analysis revealed differential genes between *B. brunneissimus* and *L. extremiorientale*, which were enriched in protein processing in the endoplasmic reticulum, as well as differential genes of the same *Boletus* species in different environments that were enriched in the ribosome pathway. The combination of metabolome and transcriptome analyses highlighted Glycine, L-Serine, and L-Aspartic acid as the key compounds responsible for the differences between the two *Boletus* species. Using the O2PLS model and Pearson’s coefficient, we identified key genes that modulate the differences in metabolites between the two species. These results have significant implications for the molecular breeding of flavor in edible mushrooms.

## 1. Introduction

Wild edible *boletus* species is highly valued for its delicious taste, rich nutrition, and various medicinal benefits and is a popular choice among consumers [1]. Research on the flavor compounds of this mushroom has become an important area of study. Non-volatile compounds, such as soluble sugars, organic acids, free amino acids, and 5′-nucleotides, are closely associated with mushroom flavor formation [2,3]. The distinct umami taste of mushrooms mainly originates from free amino acids, which can be classified into four categories: umami, sweet, bitter, and tasteless [4]. 5′-nucleotides found in mushrooms work by enhancing taste through synergistic reactions with flavoring amino acids, such as 5′-guanosine monophosphate and 5′-adenosine monophosphate [5,6]. However, research on the mechanism of flavor alteration in mushrooms remains limited, with most focusing only on changes in component content [6,7].

One of the reasons for the difficulty in studying the change mechanism of flavor substances in wild edible mushrooms may be that the artificial cultivation of most wild edible fungi is still difficult [8]. The flavor component will change significantly due to growth conditions, specifications, cooking, preservation, and storage conditions, which makes it difficult to accurately control experimental samples [9]. Researchers have attempted to overcome this obstacle by comparing different types of wild mushrooms. For example, Li et al. utilized widely targeted metabolomics to identify key taste components in two wild edible *Boletus* mushrooms [10].

*Boletus brunneissimus* (*B. brunneissimus*) and *Leccinum extremiorientale* (*L. extremiorientale*) are both wild edible *boletus* mushrooms commonly found in Yunnan markets. *B. brunneissimus* has a fruiting body with brown to dark brown color, a nearly flat cap, and dark blue flesh after injury. *L. extremiorientale*, on the other hand, has a thick and large fruiting body with an uneven cap surface that is nearly hemispherical. When it matures, it often has radial cracks, and the flesh of the fungus is light yellow, with no color change after injury [11,12]. These two *boletus* mushrooms differ in taste and flavor. Liang et al. (2016) isolated and purified three umami peptides from L. extremiorientale [13]. Wang Xiaoxiao (2017) used high-performance liquid chromatography (HPLC) to analyze the flavoring substances of *B. speciosus*, *B. auripes*, *B. edulis*, *L. extremiorientale*, and *B. brunneissimus*. The results showed that B. brunneissimus had the highest content of 5′-nucleotides [14]. However, rare studies have investigated the metabolic and molecular mechanisms underlying the differences in flavor components between these two *Boletus* mushrooms.

This study applied targeted metabolomics and transcriptome sequencing technology to investigate the differences in metabolites and gene expression between *L. extremiorientale* and *B. brunneissimus* from three distinct growing regions. The study aimed to identify the key flavor components responsible for the taste differences between the two *Boletus*, uncover the mechanisms behind these differences, and identify candidate genes that may play a role in regulating flavor components. By analyzing the molecular mechanisms of flavor component regulation in wild edible *boletus*, this research may ultimately contribute to the molecular breeding of artificially cultivated mushrooms through approaches such as genome editing and transgenic methods.

## 2. Materials and Methods

### 2.1. Mushroom Materials

To minimize the impact of environmental differences on experimental results, fruiting bodies of *B. brunneissimus* and *L. extremiorientale* were collected from three different locations in Yuxi City, Yunnan Province, at the same maturity stage [15]. The locations comprised Xiaoshiqiao Town, Hongta District, Lvzhi Town, Yimen County, and Liujie Street, Yimen County.

The experimental samples were purchased from the local market. Before purchasing, it was confirmed to the seller that the samples were collected on the local mountain and not transported from other places. Three biological replicates were collected for each *boletus* species at each site, resulting in a total of 18 experimental samples. Samples were immediately stored at −80 °C in a low-temperature refrigerator until testing. The experiment focused on the pileus, as it typically contains a higher concentration of the flavor component than the stipe [16].

### 2.2. Metabolites Extraction

To extract the samples after freeze-drying, they were first crushed in a mixer mill at 35 Hz for 45 s. The resulting material was weighed, then mixed with 700 L of extract solution, consisting of methanol and water in a ratio of 3:1, and pre-cooled to −40 °C. The internal standard was also added. Aliquots of 50 mg from each individual sample were added to Eppendorf tubes and homogenized at 35 Hz for 4 min, following a 30 s vortex, before being sonicated for 5 min in an ice water bath. This process was repeated three times. Next, the samples were shaken overnight at 4 °C and centrifuged for 15 min at 4 °C at 12,000 rpm (RCF = 13,800 (g), R = 8.6 cm). To prepare for ultrahigh-performance liquid chromatography-tandem mass spectrometry (UHPLC-MS), the supernatant was filtered through a 0.22 m microporous membrane and then diluted 20 times with an internal standard-containing solution of methanol and water (*v*:*v* = 3:1). The resulting supernatant was transferred to 2 mL glass vials. 50 L from each sample was taken and pooled as quality control(QC) samples before being stored at −80 °C [17].

### 2.3. UHPLC-MS Analysis

For UHPLC separation, an EXIONLC System from Sciex was used. The mobile phase A consisted of 1% formic acid in water, while acetonitrile served as the mobile phase B. The column temperature was fixed at 40 °C, and the autosampler was set at 4 °C. The injection volume was 2 L. For the test, a Sciex QTrap 6500+ was used, with typical ion source settings that included IonSpray Voltage at +5500/−4500 V, Curtain Gas at 35 psi, Temperature at 400 °C, Ion Source Gas 1 at 60 psi, Ion Source Gas 2 at 60 psi, and DP at 100 V. The data collected was processed using the SCIEX Analyst Work Station Software (version 1.6.3), and multiple reaction monitoring (MRM) was used. Custom R programming and database were used for peak detection and annotation based on the data matrix that comprised retention time (RT), the mass-to-charge ratio (m/z), and peak intensity. An in-house metabolite database was compared with the self-compiled R program for peak detection and annotation [18,19,20]. The metabolite profiling was conducted by Personal Biotechnology Cp. Ltd., based in Shanghai, China.

Based on the local metabolic database, the metabolites of the samples were qualitatively and quantitatively analyzed by mass spectrometry. The multi-reaction monitoring mode MRM metabolite detection Multi-peak map shows the substances that can be detected in the sample, with each color-coded mass spectrum peak representing one metabolite detected. The characteristic ions of each substance were screened by a triple four-level bar, and the signal strength (CPS) of the characteristic ions was obtained in the detector. The mass spectrometry file of the sample was opened with MultiQuant software, and the integration and correction of chromatographic peaks were carried out. The peak Area of each chromatographic peak represented the relative content of the corresponding substance. Finally, all chromatographic peak area integral data are derived and saved. In order to compare the difference in substance content of each metabolite in different samples among all detected metabolites, according to the information on metabolite retention time and peak type, the mass spectrum peak of each metabolite detected in different samples was corrected to ensure the qualitative and quantitative accuracy [21,22]. During the analysis, SIMCA software (version 16.0.2, Sartorius Stedim Data Analytics AB, Umea, Sweden) was used for principal component analysis (PCA) and orthogonal projections to latent structures—discriminant analysis (OPLS-DA) [23,24]. To evaluate the quality of the PCA, PLS-DA, and OPLS-DA models, R^2^X or R^2^Y and Q2 values were used (R^2^ represents the interpretive rate of the model, and Q^2^ represents the predictability of the model). The OPLS-DA method was used to visualize group separation and identify significantly differentially expressed metabolites (DEMs). DEMs with VIP (variable importance) > 1 and *p* < 0.05 (as determined by student t test) were considered to have undergone significant changes [25,26]. To further analyze the data, commercial databases, such as KEGG (http://www.genome.jp/kegg/ (accessed on 5 December 2022)) and MetaboAnalyst (http://metaboanalyst.ca/ (accessed on 5 December 2022)), were used for pathway enrichment analysis. For the metabolome KEGG pathway analysis, “*Agaricus bisporus* var. *bisporus* H97” was used as a comparative reference species [27]. The metabolite profiling and subsequent analyses were carried out by Personal Biotechnology Co., Ltd., based in Shanghai, China.

### 2.4. RNA Sequencing and De Novo Transcriptomic Analysis

The same batch of samples was sent separately for transcriptome and metabolome testing. To extract the total RNA from the pileus, a magnetic tissue total RNA kit was used (Product Series: DP762-T1A, TIANGEN, Beijing, China) following the manufacturer’s instructions. The extracted RNA was then evaluated for concentration, purity, and integrity using a NanoDrop spectrophotometer from Thermo Scientific. Three micrograms of RNA were utilized as the starting material for RNA sample procedures. The TruSeq RNA Sample Preparation Kit from Illumina (San Diego, CA, USA) was used to produce sequencing libraries. mRNA was initially extracted from the whole RNA using poly-T oligo-attached magnetic beads, after which fragmentation was carried out at high temperature using divalent cations in a buffer developed by Illumina. A first strand of cDNA was then produced using random oligonucleotides and Super Script II. The second strand of cDNA was synthesized using DNA Polymerase I and RNase H. The remaining overhangs were transformed into blunt ends using exonuclease/polymerase activities, after which the enzymes were eliminated. Illumina PE adapter oligonucleotides were ligated to the 3′ ends of the DNA fragments following adenylation to prepare them for hybridization. The library fragments were purified using AMPure XP technology to identify cDNA fragments between 400–500 bp in length (Beckman Coulter, Beverly, CA, USA). A 15-cycle PCR process was used to preferentially select DNA fragments with ligated adapter molecules at both ends using the Illumina PCR Primer Cocktail. The products were then purified using the AMPure XP system, and the quantification was carried out using the Agilent high-sensitivity DNA assay on the Bioanalyzer 2 system (Agilent, Inc, Santa Clara, CA, USA). Shanghai Personal Biotechnology Co., Ltd. sequenced the sequencing library on the NovaSeq 6000 platform (Illumina, San Diego, CA, USA).

To filter the raw data, Cutadapt (version 1.16) was used [28]. This experiment employed the no-reference genome analysis approach. The Trinity software (version 2.5.1) was utilized to assemble clean reads into transcripts to support further analysis [29]. Once the montaging was complete, transcript sequence files in FASTA format were obtained. The longest transcript for each gene was extracted and used as the unigene or the representative sequence of the gene. To annotate the gene function, the databases used were NR (NCBI non-redundant protein sequences), GO (Gene Ontology), KEGG (Kyoto Encyclopedia of Genes and Genome), eggNOG (evolutionary genetics of genes: non-supervised Orthologous Groups) [30], Swiss-Prot, and Pfam. RSEM (version 1.2.31) statistics were employed to compare the Read Count values for each gene for its original expression. FPKM was then used to standardize the expression. DESeq (version 1.32.0) was used for the differential analysis of gene expression (DESeq). The screening conditions for differentially expressed genes were: multiple of the expression difference |log2FoldChange| > 1, significance *p*-value < 0.05.

### 2.5. Correlation Analysis of Transcriptomic and Metabolomic Data

The analysis of the flavor metabolism mechanisms of the two edible fungi was carried out by combining the enrichment results of transcriptome differentially expressed genes and metabolome differentially expressed metabolites. To identify the differential genes that strongly correlated with the differential metabolites, the Pearson correlation coefficient (|r| > 0.8, *p* < 0.05) was used. The O2PLS model was then utilized to analyze the correlation of the two data matrices from transcriptomics and metabolomics and employed for two-way modeling and prediction in two data matrices. This approach was helpful in selecting candidate genes for future functional verification [31].

## 3. Results

### 3.1. Metabolic Profiling

The original data from the UHPLC-MS analysis consisted of 3 quality control (QC) samples and 18 experimental samples. In total, 644 peaks were extracted from both *Boletus* species, with isomers having the same peak being classified as the same class. Figure 1a shows the classification of all metabolites, with the highest number of metabolites belonging to the alkaloids (13.0%), followed by phenols (12.0%), flavonoids (10.1%), terpenoids (8.5%), and their derivative metabolite classes. Interestingly, no differences were found between the two *Boletus* species in the types of metabolites tested.

From the perspective of metabolite expression abundance accumulation, amino acid and derivatives, organic acids and derivatives, nucleotide and its derivatives, phenols, benzene, and substituted derivatives accounted for a relatively high proportion of the total abundance accumulation of the whole sample expression (Figure 1b). The proportion of these metabolite classes varied substantially depending on sampling sites and species, with the proportion of amino acids and derivatives, phenols, alkaloids, and amino acids and their derivatives varying the most. However, the proportion of organic acids and their derivatives in the expression abundance of total metabolites was relatively stable, with a small range. Therefore, the key class that influences the flavor difference between the two *Boletus* species might not be organic acids and their derivatives.

### 3.2. Analysis of the Applicability of the OPLS-DA Model to This Study

In this study, nine samples from two *Boletus* species collected from three locations were compared as a group. The Pearson correlation coefficients of metabolites in each sample are shown in Appendix A. The principal component analysis (PCA) score scatter plot shows that all data are within the 95% confidence interval (Hotelling’s T-squared ellipse). The *B. brunneissimus* and *L. extremiorientale* are located on both sides of the QC, and the *Bolete* from the same location are gathered, as can be seen in Figure 2a. These results suggest that there are significant differences in metabolites between *B. brunneissimus* and *L. extremiorientale* and that the metabolites in the fruiting bodies also differ between the different locations. Intra-group differences can help to highlight stable metabolites between each region and balance metabolites that have excessively high values in a sample, thereby optimizing the calculation of differential metabolites between groups.

After logarithmic (LOG) conversion and UV formatting of the metabolite data, the orthogonal projections to late structures-discriminant analysis (OPLS-DA) model score scatter plot of the two groups of samples is highly significant, with all samples within the 95% confidence interval (Hotelling’s T-squared ellipse, Figure 2b). The OPLS-DA modeling analysis shows that the modeling ability (R^2^Y(cum) = 0.994) and predictive ability (Q^2^(cum) = 0.769) were 99.4% and 76.9%, respectively (Figure 2c). In the OPLS-DA permutation plot (Figure 2d), all the blue Q^2^ points are lower than the original blue Q^2^ point on the right, and the regression line of the Q^2^ point intersects with the ordinate or is less than zero, indicating that the original model is stable and there is no overfitting phenomenon (R^2^Y(cum) = (0,0.92), Q^2^(cum) = (0,−0.82)). These results indicate that the OPLS-DA model is suitable for evaluating the differences between *B. brunneissimus* and *L. extremiorientale*.

### 3.3. Difference of Flavor Metabolite between B. brunneissimus and L. extremiorientale

Based on the OPLS-DA analysis, 47 metabolites were found to have significant differences (VIP > 1, *p* < 0.05) between *B. brunneissimus* and *L. extremiorientale*, which accounts for 7.3% of the total metabolites. The most significant class of differential expression metabolites (DEMs) included “amino acid and derivatives”, “nucleotide and its derivatives”, and “phenols” (Appendix A, Figure 3). Among these DEMs, 18 metabolites in the fruiting body of *L. extremiorientale* were substantially greater than those in *B. brunneissimus*, while 29 metabolites were significantly lower.

Regarding mushroom flavor metabolites, 17 free amino acids, three soluble sugars, two organic acids, and one type of 5′ nucleotide were identified. Among them, seven free amino acids and one organic acid were found to be significant metabolites (Table 1). L-Aspartic acid (umami taste), Glycine (sweet taste), D-Serine (sweet taste), L-Serine (sweet taste), L-Histidine (bitter taste), and Tryptophan (L-Tryptophan, D-Tryptophan; bitter taste) were substantially greater in the free amino acids of *L. extremiorientale* compared to *B. brunneissimus*, whereas L-Isoleucine (L-Leucine, bitter taste) and Isoleucine (bitter taste) were higher in *B. brunneissimus*. The flavor-related soluble sugar alpha-D-Glucose (D-Tagatose, sweet taste) was also found to be higher in L. extremiorientale than in *B. brunneissimus*. Moreover, the content of ethyl acetate in *B. brunneissimus* was found to be significantly higher than that in *L. extremiorientale*. Ethyl acetate is an edible spice with a fruity smell and may be the reason why *B. brunneissimus* smells more fragrant than *L. extremiorientale*.

### 3.4. Transcriptomic Difference of the Fruiting Bodies of Two Wild Edible Boletus

The KEGG database was employed to analyze the differential metabolites and 47 differential metabolites were annotated, mostly enriched in metabolic pathways (abv01100). The most significantly enriched (*p* < 0.001) pathways were “Cysteine and methionine metabolism (abv00270)”, “Glycine, serine and threonine metabolism (abv00260)”, “Biosynthesis of amino acids (abv01230)”, “ABC transporters (abv02010)”, “D-Amino acid metabolism (abv00470)”, “Cyanoamino acid metabolism (abv00460)”, “Aminoacyl-tRNA biosynthesis (abv00970)”, and “2-Oxocarboxylic acid metabolism (abv01210)”. Among these pathways, “Tryptophan metabolism (abv00380)”, “Valine, leucine and isoleucine biosynthesis (abv00280)”, “Cysteine and methionine metabolism (abv00270)”, “Glycine, serine and threonine metabolism (abv00260)”, and “Alanine, aspartate and glutamate metabolism (abv00250)” were the top 5 enriched pathways in DEMs (Figure 4).

To better understand the regulation patterns of these DEMs, we conducted RNA-seq analysis of these wild edible *Boletus*. We obtained 80.67–93.79% clean reads, and a total of 232975214 Unigenes were predicted. Among them, 17,935 differentially expressed genes (DEGs) were identified between *B. brunneissimus* and *L. extremiorientale*. The DEGs clustered in clust1 and clust4 were highly expressed in *B. brunneissimus*, while clust 5, clust 6, clust 7, and clust 8 were highly expressed in *L. extremiorientale*(Figure 5). In addition, clust1 comprised genes that were upregulated in *B. brunneissimus* while surprisingly not expressed or very little expressed in *L. extremiorientale*. These genes were most enriched in the processes of peroxisome, mRNA surveillance, and nucleocytoplasmic transport. The same expression trend appeared on clust4, and these were linked to the processes of the cell cycle, MAPK signaling pathway, and nucleocytoplasmic transport. On the contrary, genes clustered in clust 5 that were upregulated in *L. extremiorientale* exhibited the lowest expression level in *B. brunneissimus*. For clust5, a large number of genes were enriched in protein processing in the endoplasmic reticulum, cell cycle, and MAPK signaling pathways. Clust 7 and clust 8 showed similar expression patterns to clust 5 and were involved in the processes of nucleocytoplasmic transport and biosynthesis of secondary metabolites. Clust 3 showed one peak in the *B. brunneissimus* collected from Xiaoshiqiao, and the genes were enriched in the ribosome, spliceosome, and protein processing in the endoplasmic reticulum pathway. Clust 2 showed similar expression patterns mostly involved in metabolic and secondary metabolites in samples collected in Liujie, indicating these genes may be related to the microenvironment (Appendix A). The heat map of each sample’s DEGs clustering is presented in Appendix A. Further functional analysis of differential genes showed that the GO enrichment categories of protein folding and unfolded protein binding and the Protein processing in the endoplasmic reticulum pathway (ko04141) were significantly clustered (Figure 6). This result reflects the overall enrichment of transcriptome differential genes in the fruiting body of the two *Boletus* species.

### 3.5. Analysis of DEMs and DEGs in the Two Boletus Fruiting Bodies in Different Locations

The 18 samples were divided into nine groups for DEMs and DEGs analysis based on comparisons between the two *Boletus* species in the same and different locations. All nine combinations were appropriate for OPLS-DA analysis to identify differential metabolites. The differential metabolites screened by different comparison combinations are listed in Appendix A. The Venn plot (Figure 7a) and Upset plot (Figure 7b) comparing the two *Boletus* species from three regions and the same *Boletus* species from three regions showed that alpha-D-glucose (D-Tagatose) and L-carnitine were common differential metabolites between the two *Boletus* species in different regions, suggesting that the differences in these two substances between the *Boletus* species are not influenced by the environment. For the one *Boletus* species in different environment combinations, hypoxanthine, L-alanine (DL-alanine), and salsolinol were the common differential metabolites of *B. brunneissimus* and *L. extremiorientale*, respectively. These three peaks are most susceptible to environmental impact. The results of the differential metabolites analysis of *Boletus* fruiting bodies in different environments indicate that the differential metabolites of the two *Boletus* species’ fruiting bodies vary between environments.

The DEGs screened by different comparison combinations are listed in Appendix A. For the comparison of two *Boletus* species in the same region, the findings of enrichment analysis after KEGG annotation of DEGs were consistent with the prior comparative combination results, and the major enrichment was in the Protein processing in the endoplasmic reticulum (ko04141) pathway. However, the GO annotation did not have any obvious common enrichment terms. For the comparison of the same *Boletus* species in different environments, the majority of KEGG-annotated DEGs were enriched in the Ribosome pathway (ko03010), and there were no obvious common enrichment terms in the GO annotation (Appendix A).

### 3.6. Correlation Analysis of Metabolomic and Transcriptomic Data

Correlation analysis between the metabolomics and transcriptomics data (comparing *B. brunneissimus* to *L. extremiorientale*, with all nine samples of each *Boletus* as a group) revealed that out of 17,935 DEGs, 4240 DEGs showed a strong correlation with 47 DEMs using the Pearson correlation coefficient (|r| > 0.8, *p* < 0.05). The KEGG annotation of these 4240 DEGs was significantly enriched for the protein processing in the endoplasmic reticulum (ko04141) pathway, the same as the result obtained from the analysis of all the DEGs. However, the GO annotation was significantly enriched for binding and protein binding terms, which differed from the result obtained from the analysis of all the DEGs (Appendix A).

The regulatory network analysis of differential metabolites revealed that the key differential metabolites of *B. brunneissimus* and *L. extremiorientale* are Glycine, L-Serine, and L-Aspartic acid. The differential metabolites produced by these three amino acid metabolism pathways account for 38% of the total differential metabolites. Enzymes related to DEMs in the pathway were used to compare DEGs, and it was observed that some enzymes that affect the formation of DEMs were not hit by any differential genes (Figure 8). Furthermore, through KEGG analysis, a large number of genes were annotated as “hypothetical protein”, indicating that the genome and transcriptome of wild *boletus* require more in-depth research.

To better explore the differentially expressed genes that affect the metabolites of the two wild *Boletus* species, we used the O2PLS model to analyze the DEGs and metabolites strongly associated with the DEMs (with the logarithm of the peak value used for analysis). The results are presented in Figure 9, and the O2PLS model better explains the association between the metabolome and transcriptome (with R^2^X = 0.89, R^2^Y = 0.692, R^2^Xcorr = 0.75, R^2^Ycorr = 0.574). The top 10 genes, such as “TRINITY-DN343297_c0_g1,” have the highest sum of loading squared values and can be used as candidate genes for further studying the interspecific differences between the two *Boletus* species (Table 2).

To investigate the key genes responsible for flavor differences between the two types of *Boletus* species, we conducted Pearson correlation analysis (|r| > 0.8, *p* < 0.05) using seven amino acid differential metabolites related to flavor (L-Aspartic acid, Glycine, D-Serine, L-Serine, L-Histidine, Tryptophan, and L-Isoleucine) and differential genes, and a total of 276 differential genes were identified. Subsequently, using the seven DEMs and 276 strongly associated DEGs for O2PLS analysis (with R^2^X = 0.95, R^2^Y = 0.82, R^2^Xcorr = 0.79, R^2^Ycorr = 0.82), we selected ten genes, including “TRINITY_DN455_c1_g1,” with the highest loading square value. These candidate genes can be used to further study the differences in the flavor of the two *Boletus* species (Table 2).

**Table 1 foods-12-02728-t001:** Statistics of accumulating free amino acids, soluble sugars, organic acids, 5′-nucleotides, and miscellaneous in the fruiting body of. “ *B. brunneissimus*”and “*L. extremiorientale*”.

Index	Compound name	Class	Taste	Peak Area		VIP	*p*-Value	*B. brunneissimus* vs *L. extremiorientale*
				*B. brunneissimus*	*L. extremiorientale*			
1	L-Aspartic acid	Free amino acids	Umami taste (MSG-like)	394,035.09 ± 192,939.3	618,031.63 ± 180,516.1	1.51	0.02	UP
2	D-Aspartic acid	Free amino acids	Umami taste (MSG-like)	433,700.93 ± 232,571.85	632,223.46 ± 316,790.67	1.03	0.13	-
3	L-Glutamic acid	Free amino acids	Umami taste (MSG-like)	6,201,893.95 ± 3,509,410.93	6,921,258.43 ± 1,860,205.86	0.62	0.76	-
4	L-Alanine; DL-Alanine	Free amino acids	Sweet taste	658,758.14 ± 413,661.32	1,382,503.87 ± 922,614.21	0.06	0.22	-
5	Glycine	Free amino acids	Sweet taste	48,034.62 ± 15,782.87	114,075.09 ± 16,451.79	2.19	0	UP
6	D-Proline	Free amino acids	Sweet taste	8,407,880.54 ± 583,9726	11,620,670.17 ± 4,494,533.82	0.99	0.3	-
7	D-Serine	Free amino acids	Sweet taste	114,362.85 ± 68,449.37	316,275.73 ± 147,831.59	1.82	0	UP
8	L-Serine	Free amino acids	Sweet taste	111,179.47 ± 70,914.14	316,275.73 ± 147,831.59	1.81	0	UP
9	L-Threonine	Free amino acids	Sweet taste	576,328.69 ± 392,016.92	692,317.82 ± 153,705.11	0.88	0.44	-
10	L-Histidine	Free amino acids	Bitter taste	5,495,343.35 ± 2,751,086.31	9,208,274.72 ± 3,538,910.94	1.47	0.03	UP
11	L-Isoleucine; L-Leucine	Free amino acids	Bitter taste	378,672.76 ± 218,557.65	188,871.1 ± 108,314.07	1.3	0.46	DOWN
12	Isoleucine	Free amino acids	Bitter taste	400,448.58 ± 258,714.71	176,708.49 ± 108,579.05	1.43	0.03	DOWN
13	L-Methionine	Free amino acids	Bitter taste	106,123.63 ± 97,300.41	47,486.83 ± 42,604.14	1.28	0.13	-
14	L-Phenylalanine;D-(+)-Phenylalanine; DL-Phenylalanine	Free amino acids	Bitter taste	21,820,440.32 ± 12,846,566.66	19,022,942.97 ± 13,038,839.53	0.47	0.61	-
15	Tryptophan(L-Tryptophan; D-Tryptophan)	Free amino acids	Bitter taste	17,996.51 ± 16,094.24	53,685.79 ± 18,460.91	1.79	0	UP
16	L-Valine	Free amino acids	Bitter taste	409,058.87 ± 139,076.01	401,702.04 ± 71,429.27	0.05	0.69	-
17	L-Arginine	Free amino acids	Bitter taste	12,571,655.39 ± 1,273,7601.53	21,443,582.43 ± 13,314,464.99	1.2	0.2	-
18	L-Lysine;L-Glutamine	Free amino acids	Tasteless	18,074,414.78 ± 7,399,759.51	23,351,362.08 ± 8,585,564.65	0.9	0.2	-
19	L-Tyrosine	Free amino acids	Tasteless	1,958,434.87 ± 1,135,802.84	2,938,080.2 ± 1,187,109.94	1.16	0.19	-
20	Inositol	Soluble sugars; Polyols	Sweet taste	3100.23 ± 1151.09	2397.99 ± 1140.6	1	0.16	-
21	D-Xylulose	Soluble sugars; Reducing sugars	Sweet taste	73,430.52 ± 62,704.61	27,330.13 ± 25,050.39	-	-	-
22	alpha-D-Glucose; D-Tagatose	Soluble sugars; Reducing sugars	Sweet taste	301,634.92 ± 128,026.33	83,124.42 ± 34,406.66	2.25	0	UP
23	Beta-D-Glucose; alpha-D-Glucose	Soluble sugars; Reducing sugars	Sweet taste	3648.68 ± 1313.4	5118.91 ± 2292.21	-	-	-
24	Sucrose	Soluble sugars; Non-reducing sugars	Sweet taste	92,833.83 ± 62,857.55	349,897.24 ± 510,820.3	-	-	-
25	L-Malic acid	Organic acids	sour taste	2013.11 ± 1300.98	1580.5 ± 1538.06	-	-	-
26	Ascorbic acid	Organic acids	sour taste	7771.71 ± 2522.87	6542.41 ± 2207.86	0.83	0.2	-
27	Adenosine 5′-monophosphate	5′-nucleotides	-	1,703,525.03 ± 710,544.37	2,947,050.82 ± 3,418,622.63	0.12	0.32	-
28	Ethyl caproate	Miscellaneous	-	64,844.26 ± 21,913.66	41,651.35 ± 22,101.1	1.48	0.03	DOWN

**Table 2 foods-12-02728-t002:** The top 10 DEGs associated with DEMs by O2PLS analysis.

The top 10 DEGs Associated with All DEMs by O2PLS Analysis				
DEGs	Loading_1	Loading_2	Loading^2 1^	NR Annotation of DEGs
TRINITY_DN343297_c0_g1	−0.011662098	−0.031064851	0.00110103	gi|1000864728|gb|KXN90516.1| Vesicle transport protein SFT2B [*Leucoagaricus* sp. SymC.cos]
TRINITY_DN15220_c0_g1	−0.013141608	−0.030371976	0.001095159	gi|751055722|gb|KIK90884.1| hypothetical protein PAXRUDRAFT_831297 [*Paxillus rubicundulus* Ve08.2h10]
TRINITY_DN7287_c2_g1	−0.013351728	−0.030237734	0.001092589	gi|914258129|gb|KNZ73472.1| WD repeat-containing protein 5 [*Termitomyces* sp. J132]
TRINITY_DN50170_c0_g1	−0.011257006	−0.030734214	0.001071312	gi|750997013|gb|KIK39305.1| hypothetical protein CY34DRAFT_89493 [*Suillus luteus* UH-Slu-Lm8-n1]
TRINITY_DN18074_c0_g1	−0.011755046	−0.030544198	0.001071129	gi|751064906|gb|KIK99469.1| hypothetical protein PAXRUDRAFT_131997 [*Paxillus rubicundulus* Ve08.2h10]
TRINITY_DN3215_c0_g3	−0.012219336	−0.03025251	0.001064527	gi|749895491|gb|KIJ63208.1| hypothetical protein HYDPIDRAFT_29895 [*Hydnomerulius pinastri* MD-312]
TRINITY_DN262_c0_g2	−0.014473261	−0.029082473	0.001055266	- ^2^
TRINITY_DN700_c2_g1	−0.014895131	−0.028608579	0.001040316	gi|749834671|gb|KIJ12323.1| hypothetical protein PAXINDRAFT_156963 [*Paxillus involutus* ATCC 200175]
TRINITY_DN10222_c0_g2	−0.013377283	−0.028754748	0.001005787	gi|751062861|gb|KIK97499.1| hypothetical protein PAXRUDRAFT_136196 [*Paxillus rubicundulus* Ve08.2h10]
TRINITY_DN5782_c0_g1	0.019328067	0.024823556	0.000989783	gi|749846545|gb|KIJ21256.1| hypothetical protein PAXINDRAFT_64023 [*Paxillus involutus* ATCC 200175]
The top 10 DEGs associated with flavor related DEMs by O2PLS analysis				
DEGs	Loading_1	Loading_2	Loading^2 1^	NR annotation of DEGs
TRINITY_DN455_c1_g1	0.040883277	0.149953165	0.024157394	gi|1035381610|gb|OAX38220.1| hypothetical protein K503DRAFT_850105 [*Rhizopogon vinicolor* AM-OR11-026]
TRINITY_DN24602_c0_g1	0.044125314	0.147860791	0.023809857	gi|749890950|gb|KIJ58942.1| hypothetical protein HYDPIDRAFT_162871 [*Hydnomerulius pinastri* MD-312]
TRINITY_DN6766_c0_g1	0.043559178	0.147316192	0.023599462	gi|628854451|ref|XP_007774347.1| hypothetical protein CONPUDRAFT_112209 [*Coniophora puteana* RWD-64-598 SS2]
TRINITY_DN19391_c0_g1	0.043673434	0.146674332	0.023420729	gi|749903138|gb|KIJ70603.1| hypothetical protein HYDPIDRAFT_23695 [*Hydnomerulius pinastri* MD-312]
TRINITY_DN59157_c0_g1	0.04454385	0.146046036	0.023313599	gi|763724697|gb|KJA22132.1| hypothetical protein HYPSUDRAFT_41267 [*Hypholoma sublateritium* FD-334 SS-4]
TRINITY_DN5293_c0_g1	0.043968374	0.145400401	0.023074494	gi|749841699|gb|KIJ16704.1| acetolactate synthase [*Paxillus involutus* ATCC 200175]
TRINITY_DN15287_c0_g1	0.047024185	0.142239221	0.02244327	gi|749892543|gb|KIJ60404.1| hypothetical protein HYDPIDRAFT_177471 [*Hydnomerulius pinastri* MD-312]
TRINITY_DN3921_c0_g1	0.046666069	0.141663398	0.02224624	gi|749830483|gb|KIJ08773.1| glycoside hydrolase family 89 protein [*Paxillus involutus* ATCC 200175]
TRINITY_DN130487_c0_g1	0.04132318	0.142957832	0.022144547	gi|749845521|gb|KIJ20338.1| hypothetical protein PAXINDRAFT_108316 [*Paxillus involutus* ATCC 200175]
TRINITY_DN9716_c0_g1	0.046452929	0.141217718	0.022100318	gi|749901809|gb|KIJ69300.1| hypothetical protein HYDPIDRAFT_145192 [*Hydnomerulius pinastri* MD-312]

Note: ^1^. Loading^2^ = (Loading_1)^2^ + (Loading_2)^2^; ^2^. “-“represents not being annotated.

## 4. Discussion

Edible mushrooms offer a delicious and nutritious addition to one’s diet, thanks to their high levels of various secondary metabolites, such as phenolic compounds, polyketones, terpenoids, and steroids. These compounds have numerous beneficial health effects, such as immune regulation, lipid-lowering, and anti-tumor properties. Additionally, consumers enjoy their rich flavor. This experiment utilized UHPLC-MS technology to perform metabolomics analysis on the fruiting bodies of *B. brunneissimus* and *L. extremiorientale*, both of which possess strong flavor differences. Of 644 annotated compounds, alkaloids, phenols, flavonoids, and terpenoids were prominently identified in both wild *Boletus* species, indicating that they are nutrient-rich and have high utilization value.

Through OPLS-DA analysis, 47 differential metabolites were identified, primarily in amino acid metabolic pathways. Transcriptome analysis found differential genes in Protein processing in the endoplasmic reticulum (ko04141) pathway for *B. brunneissimus* and *L. extremiorientale*, while the differential genes for the same wild *Boletus* species were mainly enriched in the Ribosome pathway (ko03010). A total of 8 differential metabolites relating to soluble sugars, organic acids, free amino acids, and 5′ -nucleotides were identified between *B. brunneissimus* and *L. extremiorientale*. Joint analysis of the transcriptome and metabolome showed that Glycine, L-Serine, and L-Aspartic acid were key compounds accounting for the metabolic differences between *B. brunneissimus* and *L. extremiorientale*. Finally, the O2PLS model combined with the Pearson coefficient was used to conduct further research on the screening of DEGs associated with DEMs.

Mushroom flavor is complex and mainly formed by the aliphatic components, amino acids and flavorful nucleotides, such as 1-octen-3-ol,2-octen1-ol, 3-octanol, 1-octanol, 3-octanone, 5′-GMP and glutamic acid [32]. These compounds probably form a basal flavor and may be modified to create a more complex flavor in other species, such as Lentinus edodes [33]. Lipidtemns like terpenoids, 3-(methylthio)propanal, and N-heterocyclic compounds contribute to individual odors between different mushroom species. Ref. [34] is very common in most mushroom species. Aldehydes, esters, alcohols, acids, pyrazines, ketones, and phenols; 1-octen-3-ol and 2,5-dimethylpyrazine were enriched in boletus varieties, while 3-(methylthio) propionaldehyde and 2,6-dimethylpyrazine were the main compounds in B. edulis. Benzaldehyde, 3-methyl-1-butanol, Acetoin, 2-pentylfuran, 2-heptenal, 3-octanone and ethyl tiglate composite flavor and fragrance ingredient in B. pinophilus. [35], in this study, we focus on the flavored nucleotide and amino acid and found that the key differential metabolites of *B. brunneissimus* and *L. extremiorientale* are Glycine, L-Serine, and L-Aspartic acid. The differential metabolites produced by these three amino acid metabolism pathways account for 38% of the total differential metabolites. One of the key difficulties in studying the metabolites of wild *Boletus* is the lack of control. This is in contrast to cultivated edible mushrooms, where samples from different varieties of the same species can be more easily compared to studying differential metabolites and genes. In the case of wild *Boletus*, the composition and content of metabolites are influenced by various growth factors, making it challenging to collect samples and guarantee differences. To overcome this challenge, this study utilized three sets of samples from different growth environments to create intra-group differences, thereby mitigating errors and focusing on differences in species characteristics. The experimental results showed that the different metabolites screened from different locations varied and that expanding the sample size could yield more accurate results. By conducting research in this way, we can gain a better understanding of the unique characteristics of wild *Boletus* and their metabolites.

This study involved screening out the genes related to flavor differences between *B. brunneissimus* and *L. extremiorientale* through combined analysis of the metabolome and transcriptome. Gene annotation revealed that a significant number of genes were annotated as “hypothetical protein”, suggesting that there is a wealth of untapped genetic resources in wild *Boletus*. From a breeding perspective, these genes from wild fungi can be used in the breeding of artificial fungi through large-scale fungal genetic transformation technology. The wide variety of wild *Boletus* provides an extensive gene resource pool for improving artificial mushroom varieties [36]. To study the molecular mechanism of *boletus* flavor substances regulation, it is possible to identify key regulatory genes, clone and study their functions through yeast genetic transformation, and ultimately employ genome editing, transgenic, and other technologies to improve the flavor of artificial mushrooms. Such research has important breeding implications for enhancing the flavor of artificial mushrooms.

## Figures and Tables

**Figure 1 foods-12-02728-f001:**
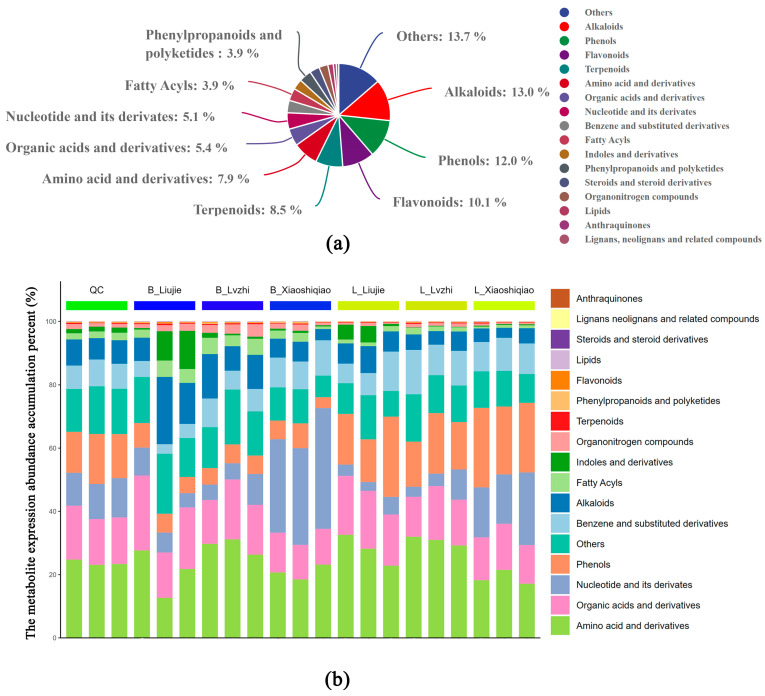
The classification of all metabolites and the metabolite expression abundance accumulation of the two *Boletus* species. (**a**) The classification of all metabolites of the two *Boletus* species. (**b**) The metabolite expression abundance accumulation of the two *Boletus* species. QC: Quality control samples. B_(Liujie, Lvzhi, Xiaoshiqiao): Collected from different locations of the *B. brunneissimus* sample. L_(Liujie, Lvzhi, Xiaoshiqiao): Collected from different locations of the *L. extremiorientale* sample.

**Figure 2 foods-12-02728-f002:**
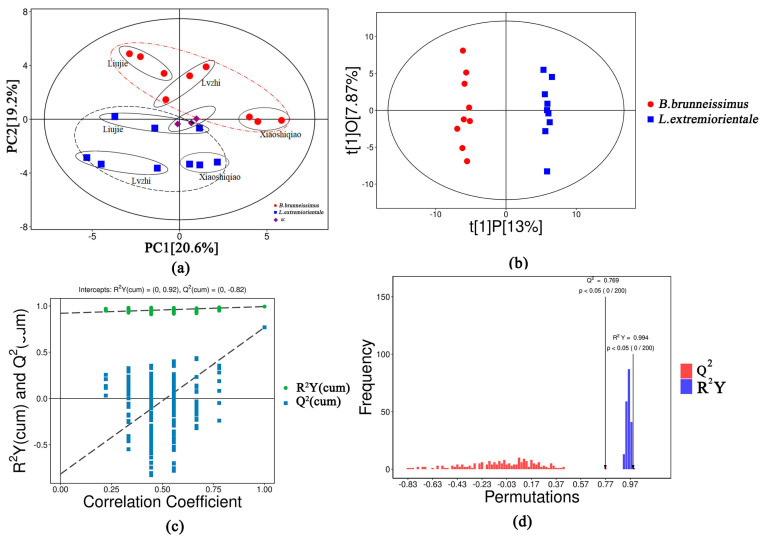
Analysis of the applicability of the OPLS-DA model to this study. (**a**) The principal component analysis (PCA) of samples from two *Boletus* species. (**b**) PCA of OPLS-DA model. (**c**) The correlation coefficient analysis of the OPLS-DA model. (**d**) The permutations analysis of the OPLS-DA model.

**Figure 3 foods-12-02728-f003:**
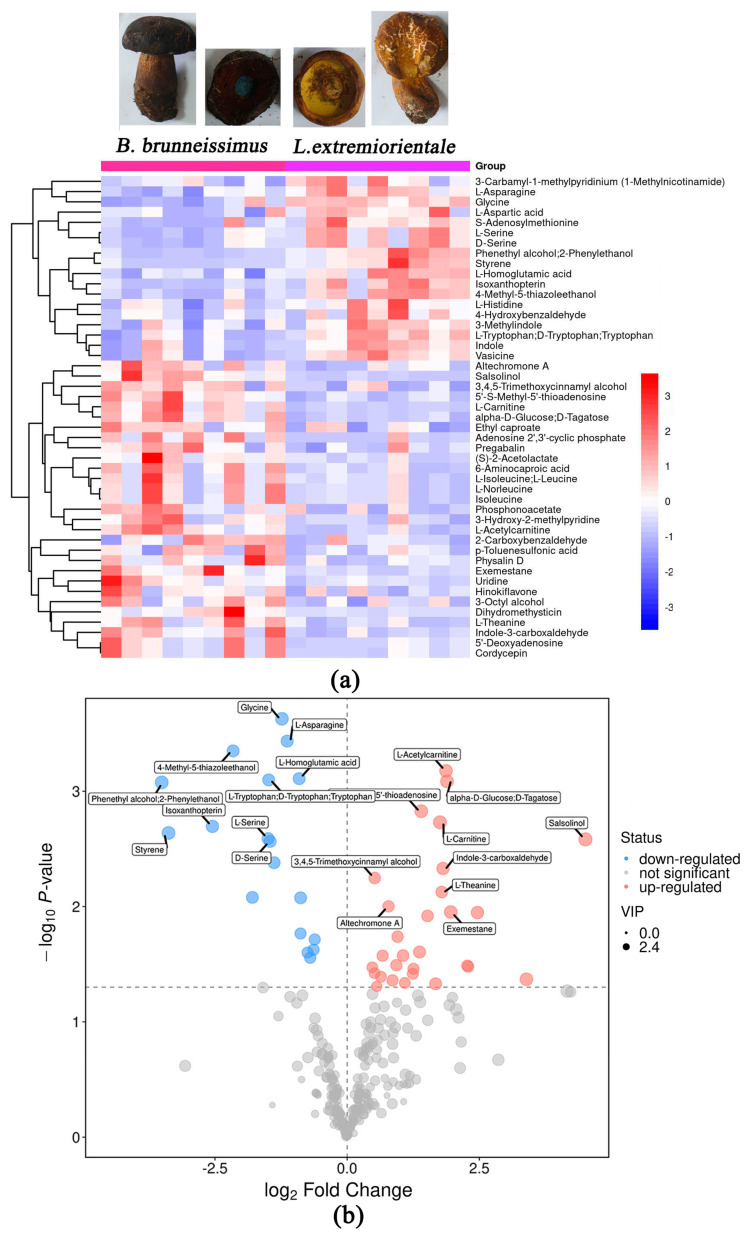
The heatmap plot (**a**) and the volcano plot (**b**) of differentially expressed metabolites.

**Figure 4 foods-12-02728-f004:**
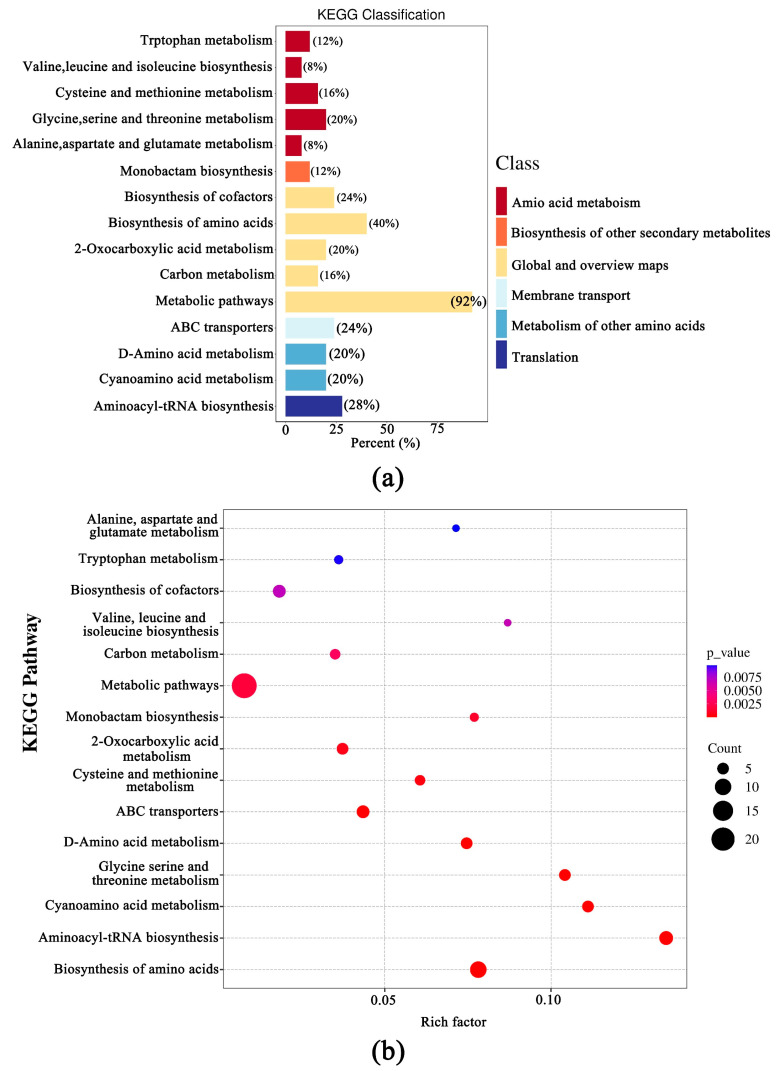
The KEGG pathway rich plot (**a**) and the bubble plot (**b**) of differentially expressed metabolites.

**Figure 5 foods-12-02728-f005:**
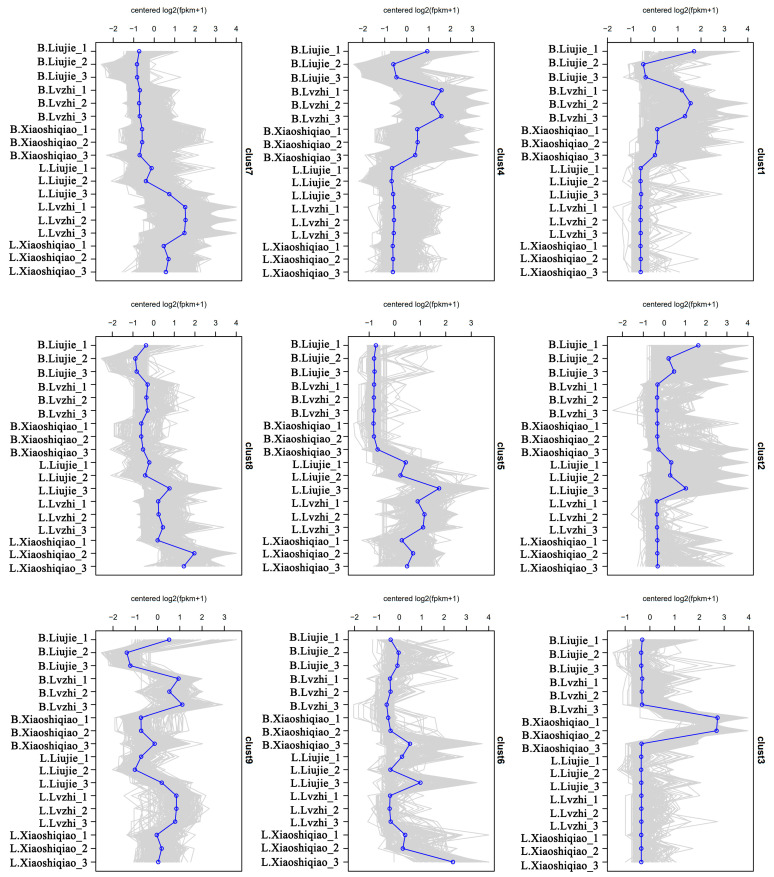
The cluster of DEGs of 18 wild edible *Boletus* samples. B.(Liujie, Lvzhi, Xiaoshiqiao): Collected from different locations of the *B.brunneissimus* sample. L.(Liujie, Lvzhi, Xiaoshiqiao): Collected from different locations of the *L. extremiorientale* sample.

**Figure 6 foods-12-02728-f006:**
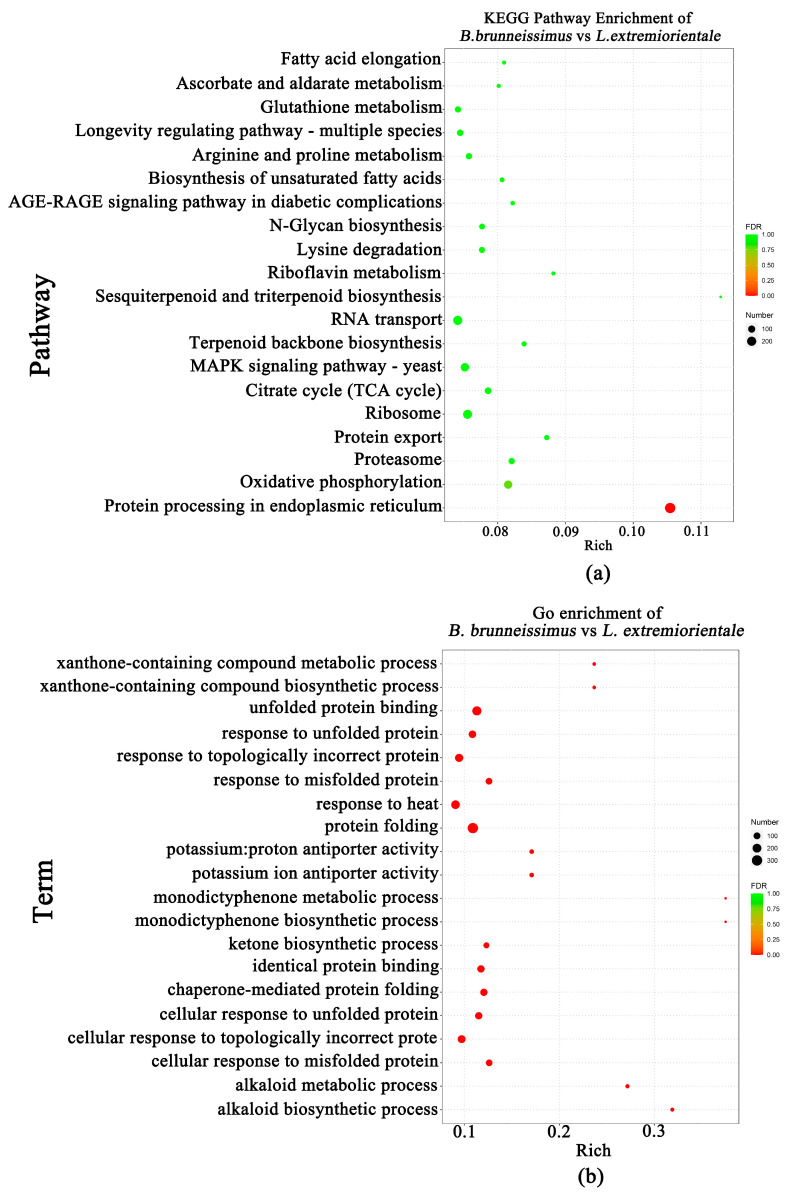
The KEGG pathway enrichment (**a**) and Go term (**b**) enrichment of *B. brunneissimus* vs. *L. extremiorientale*.

**Figure 7 foods-12-02728-f007:**
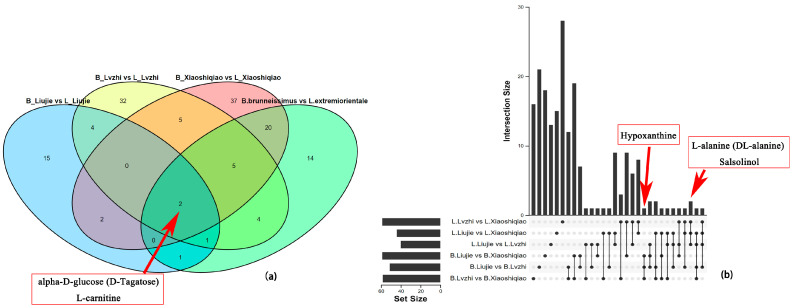
The Venn plot (**a**) and Upset(**b**) plot of different comparison combinations of *B. brunneissimus vs L. extremiorientale*. B_(Liujie, Lvzhi, Xiaoshiqiao): Collected from different locations of the *B. Brunneissimus sample*. L_(Liujie, Lvzhi, Xiaoshiqiao): Collected from different locations of the *L. extremiorientale* sample.

**Figure 8 foods-12-02728-f008:**
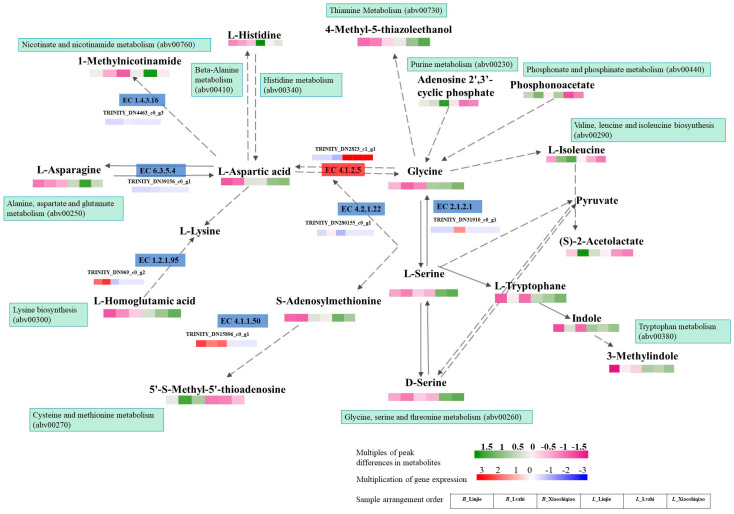
Correlation analysis between the metabolomics and transcriptomics data. The solid line represents the direct conversion of differential compounds; The dashed line represents the indirect transformation of differential compounds.

**Figure 9 foods-12-02728-f009:**
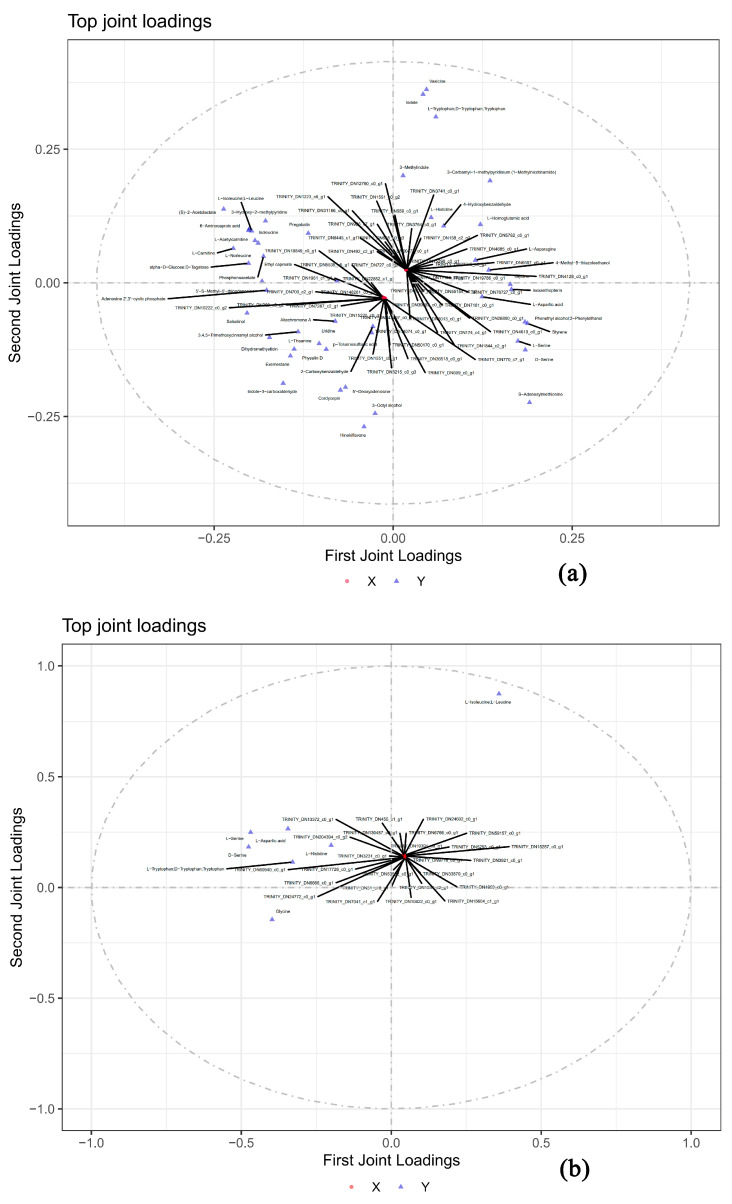
Analysis results of O2PLS model. (**a**) The relationship between all DEMs and all DEGs. (**b**) The relationship between flavor-related DEMs and all DEGs.

## Data Availability

Data are contained within the article or Appendix A.

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
