# Peer review of "Insights into the Flavor Differentiation between Two Wild Edible Boletus Species through Metabolomic and Transcriptomic Analyses"

_foods, 2023, doi:10.3390/foods12142728_

Round 1

Reviewer 1 Report

Thank you for allowing me to review your article investigating the transcriptomic and metabolomic differences between two wild edible Boletus species.

Your paper was very thorough, and acknowledged some of the limitations of the work.

Please could you define exactly how you obtained the samples. Were they growing in the wild, or purchased? Is there any data specifically looking at differences between each of the 3 samples from the same location? You mention within site differences, and this could be described/discussed more clearly... You mention that 'the results of the differential metabolites analysis of Boletus fruiting bodies in different environments indicate that the differential metabolites of the two Boletus 287 species' fruiting bodies vary between environments", but how much variability occurs within environments?

The 'clusters' referred to in section 3.4 could be better defined/explained.

Some of the figures have a lot of details that are very hard to see. For example on Fig. 4a, even at a magnification of 200% I still struggled to read the figure labels. Increasing the size of some of the figures would help. Likewise, Table 1. was tricky to read. There is a lot of information present, and as you go down the table, it is hard to keep track of what each of the table columns are.

Language was acceptable.

Reviewer 2 Report

This study deals with the metabolomics and transcriptomics analyses of two Boletus spp. Standard methodologies were used but some parts lack clarity. This manuscript requires substantial revision.

1. The title, which includes "flavour differentiation mechanisms", do not match the content of the manuscript.

2. The authors stated they use samples at same maturity stage (line 77) but there is no mention on how the stages were defined. In addition, any authentication of the samples was performed?

3. The description of samples used in this study is rather confusing, such as "three duplicate samples", "originating from a distinct individual"(=mushroom"?). In the PCA plots, there are 3 replicates for each location?

4. The number of samples is on the low side. Most metabolomic studies use up to 5/6 samples. What didn't the authors include more replicates since the supply for cultivated mushrooms is often not an issue?

5. Did the authors used the same mushroom samples for both metabolomics and transcriptomics studies?

6. The manner by which the identification of the compounds was achieved was not described clearly. Accurate identification of the compounds is crucial (refer below).

7. Some of the claims could have been supported with additional evidence. For instance, " Ethyl acetate is an edible spice with a fruity smell..." (line 243)? Have any previous researchers reported the presence of ethyl acetate in mushrooms? 

8. The discussion section is too short. There is a lack of comparison of the findings of this study with previous studies. To what extent the objectives were achieved?

9. Lines 406-408 touched on artificial bacteria? Is this even relevant to this work?

10. This manuscript requires thorough language editing.

Requires editing
